# Surgical Instrument Signaling Gesture Recognition Using Surface Electromyography Signals

**DOI:** 10.3390/s23136233

**Published:** 2023-07-07

**Authors:** Melissa La Banca Freitas, José Jair Alves Mendes, Thiago Simões Dias, Hugo Valadares Siqueira, Sergio Luiz Stevan

**Affiliations:** 1Graduate Program in Electrical Engineering (PPGEE), Federal University of Technology–Paraná (UTFPR), Ponta Grossa 84017-220, PR, Brazil; melissa.1995@alunos.utfpr.edu.br (M.L.B.F.); hugosiqueira@utfpr.edu.br (H.V.S.); 2Graduate Program in Electrical Engineering and Industrial Informatics (CPGEI), Federal University of Technology–Paraná (UTFPR), Curitiba 80230-901, PR, Brazil; jjjunior@utfpr.edu.br (J.J.A.M.J.); tsdias@utfpr.edu.br (T.S.D.)

**Keywords:** pattern recognition, surface electromyography, signal processing, automatic segmentation, ensemble classification, telesurgery, robotic surgery

## Abstract

Surgical Instrument Signaling (SIS) is compounded by specific hand gestures used by the communication between the surgeon and surgical instrumentator. With SIS, the surgeon executes signals representing determined instruments in order to avoid error and communication failures. This work presented the feasibility of an SIS gesture recognition system using surface electromyographic (sEMG) signals acquired from the Myo armband, aiming to build a processing routine that aids telesurgery or robotic surgery applications. Unlike other works that use up to 10 gestures to represent and classify SIS gestures, a database with 14 selected gestures for SIS was recorded from 10 volunteers, with 30 repetitions per user. Segmentation, feature extraction, feature selection, and classification were performed, and several parameters were evaluated. These steps were performed by taking into account a wearable application, for which the complexity of pattern recognition algorithms is crucial. The system was tested offline and verified as to its contribution for all databases and each volunteer individually. An automatic segmentation algorithm was applied to identify the muscle activation; thus, 13 feature sets and 6 classifiers were tested. Moreover, 2 ensemble techniques aided in separating the sEMG signals into the 14 SIS gestures. Accuracy of 76% was obtained for the Support Vector Machine classifier for all databases and 88% for analyzing the volunteers individually. The system was demonstrated to be suitable for SIS gesture recognition using sEMG signals for wearable applications.

## 1. Introduction

Efficient communication between surgeon and their team is essential to a successful surgical intervention [1,2]. Communication between the surgeon and the surgical instrumentator, who is responsible for organizing the instrument tray and providing the instruments when requested, is made using verbal and surgical instrument signaling. In surgical instrument signaling (SIS), the surgeon executes hand gestures representing determined instruments, which is used to avoid errors and communication failures [3,4].

Surgical instrumentation is related to manipulating instruments used during surgical interventions, examinations, treatments, and dressings [3,5,6]. It is one of the most important areas in surgical procedures, as these instruments can be considered an extension of the surgeon’s hands. Depending on the type of surgery (cardiothoracic surgery, plastic surgery, neurosurgery, etc.), different types of instruments are used. For this reason, there are many existing instruments [3,4].

However, sometimes the medical instrumentator is not available for a surgical procedure. In these cases, the surgery is often postponed. Therefore, inserting technological tools could aid the surgical instrumentation process, such as surgical robots [7]. This type of application requires systems which are able to recognize gestures and requests made by the physician during the surgery. The gesture recognition for SIS gestures can reduce the instrumentator dependence, which could be an advantage. Note that the presence of the instrumentator typically implies additional costs for surgical procedures. Moreover, in telesurgery and robotic surgery, the possibility of errors could be avoided because of the absence of instrumentator tiredness, inattention, and fatigue.

In this sense, this application can benefit from using technological and/or assistive tools to promote the better performance of surgical procedures, whether related to robotics or telesurgery [8,9,10].

Usually, SIS systems use pattern recognition based on images or videos [11,12,13], and image systems are highly dependent on environmental factors, such as lighting, high contrast, shadows, and background color [14]. As the operation room presents high lighting and most equipment is metallic, it can be difficult to operate the classification system. On the other hand, systems based on speech recognition can be influenced by the background noise of the devices in the surgical environment. Thus, the use of wearable devices for gesture recognition that work independently of the aforementioned conditions could be a redundancy system, such as gloves [15] or armbands [16].

Some techniques have been developed based on robots that aid physicians in SIS systems. Jacob et al. [2,17] developed a robot (Gestonurse) that recognized five gestures via Kinect sensor, and each gesture was related to a surgical instrument. The instruments recognized were a scalpel, scissors, retractor, forceps, and hemostat. The authors noted that, even with high accuracy (about 95%), the robot was 0.83 s slower than the human response. The authors improved this system by inserting speech recognition and anti-collision routines, increasing the number of gestures and the accuracy, and decreasing the robot response. However, the gestures used were not the signals which are typical for surgeons during the interventions [18,19]. Similarly, using speech recognition, Perez-Vidal et al. [20] developed a robotic device able to distinguish 27 surgical instruments and 82 spoken instructions.

In this perspective, the recognition of surgical tray instruments was developed by Zhou and Wachs [21], using computer vision to be applied to robotic devices. Computer vision was also used by Le and Pham [11] to identify surgical instruments using a Kinect to reduce delays and failures. Five gestures were classified using symbols to identify each instrument.

Works involving the acquisition of signals from wearable sensors are becoming more common in these applications. Among these techniques, surface electromyography (sEMG) has gained ground for performing SIS recognition. For example, Bieck et al. [16], in 2019, proposed an SIS recognition system using the commercial Myo armband to capture sEMG signals from the forearm during the handling of five surgical instruments: straight clamp, forceps, perforator, bone punch, and sharp spoon. Hit rates above 90% were achieved for some training conditions. In the work from Qi et al. [22] developed in 2022, gesture recognition of numbers from 1 to 10 was developed in such a way as to improve the human–machine relationship for robotic telesurgery systems. Using sEMG and deep learning classifiers, authors reached hit rates above 80% for the selected gestures [22]. These works demonstrate the possibility of integration between sEMG devices and SIS gestures, even though this perspective is still scarcely present in the literature.

The sEMG signal is acquired through voluntary contractions of skeletal muscles and can be obtained on the skin surface with non-invasive electrodes [23,24]. The sEMG signal could present some advantageous features in this approach. Firstly, sEMG is less sensitive to noise due to external factors, such as ambient lighting and sounds, which occurs with devices based on image acquisition or speech [25,26,27]. Furthermore, it can be easily detected when there is muscle activation, not requiring expensive pre-processing routines to identify a new gesture, as occurs with the treatment of image systems [26]. As it is a signal from muscle activation, much information can be taken, not just those related to the recognition of gestures. On the other hand, some limitations of sEMG acquisition include muscle crosstalk (from muscle groups near to the place of interest), uncorrected electrode placement, muscle fatigue, and interference from other biological signals, such as electrocardiograms [28,29]. However, these can be minimized using wearable acquisition devices.

Considering these issues, this work proposes a system for SIS recognition based on sEMG signals using a wearable device (Myo™ armband). In this work, 14 gestures are considered, i.e., a larger set than previous work [22]. We present the feasibility of using only sEMG signals in the pattern recognition process to aid surgical procedures. Our study was not designed to substitute one technology for another, but to demonstrate how a wearable sensor with machine learning frameworks can be applied to solve this problem, which can act as a redundancy system. In addition, the segmentation, feature extraction and selection, and classification steps are explored to identify these signals. Signal processing stages were performed, taking into account the use of a wearable device, in which the complexity of pattern recognition algorithms is crucial.

This paper is organized as follows: Section 2 presents an overview of the SIS; the methodology, which includes signal processing steps and analysis, is described in Section 3; the results are presented in Section 4; and Section 5 and Section 6 provide the discussion and conclusions, respectively.

## 2. Surgical Instrument Signaling

Surgical Instrument Signaling (SIS) is a specific language based on the gesture set adopted in the surgical environment to aid communication in maneuvers performed during surgical procedures. Using the SIS, the physician transmits the information about the instrument that he/she needs, aiding in decreasing the time of surgery. It is crucial when there is a high level of noise in the surgery room or there are difficulties in verbalization, to increase concentration and improve communication during the surgical procedure [30,31].

Furthermore, not all instruments have specific signaling; however, the main instruments have signaling that is commonly used in the surgical environment. Several instruments can be used in surgery, and several SIS gestures can be applied in surgery. Thus, a set of SIS gestures was selected; the set used in this work is presented in Figure 1. One of the criteria for choosing the gestures was using only one hand for their performance. In total, 14 selected gestures for SIS were chosen, unlike similar previous works that used up to 10 gestures to represent or classify SIS gestures. The following signals were used [3,4,32]:Compress (Figure 1-1): Hand is flattened with fingers together and the palmar surface facing upwards;Medical Thread on a Spool (Figure 1-2): Palm’s hand faces upwards with fingers semi-flexed;Medical Thread Loose (Figure 1-3): Hand is palm down with fingers semi-flexed;Plier Backhaus (Figure 1-4): Ring, middle, and index fingers are flexed while the thumb interposes the index and middle fingers;Hemostatic Forceps (Figure 1-5): Index and middle fingers are crossed with the palm side of the hand facing down;Kelly Hemostatic Forceps (Figure 1-6): Ring and little fingers are flexed, and the other fingers are extended;Farabeuf Retractor (Figure 1-7): Index finger is semi-flexed, and the other fingers are flexed; the hand moves similarly to handling the instrument;Bistouri (Figure 1-8): All fingers are semi-flexed and gathered at the tips, performing a pendular movement, similar to the movement performed when handling the instrument;Needle Holder (Figure 1-9): Index, middle, ring, and pinkie fingers are semi-flexed, and the thumb is partially flexed on the opposite side; the hand performs small rotational movements;Valve Doyen (Figure 1-10): Hand moves with all fingers together; the fingers are stretched out and at right angles to the rest of the hand;Allis Clamp (Figure 1-11): Thumb and index fingers are semi-flexed, making an opening and closing movement with the thumb holding the index finger, while the other fingers remain flexed;Anatomical Tweezers (Figure 1-12): Thumb and index fingers are extended, making an approach and removal movement, while the other fingers remain flexed;Rat’s Tooth Forceps (Figure 1-13): Thumb and index fingers are semi-flexed (making an opening and closing movement with the ends of the index finger and thumb touching each other), and the other fingers remain flexed;Scissors (Figure 1-14): Index and middle fingers are kept extended.

## 3. Methodology

After the delimitation of the gestures, the data acquisition process started. Figure 2 presents the main flow of data acquisition. In this work, the sEMG signal acquisition was performed using the commercial Myo™ armband (Figure 2a). This wearable device has eight sEMG acquisition channels, a nine-axis inertial motion unity (three-axis magnetometer, accelerometer, and gyroscope), and a vibration motor acting with pre-determined events. An ARM Cortex M4 (MK22FNIM) was used to process the data, with a 200 samples/second sampling rate and 8-bit resolution, and the data were transmitted using Bluetooth protocol [33].

For data acquisition, the Myo™ device was placed on the right forearm, using as reference the third part of the total length of the forearm (Figure 2a). Electrodes were placed in a way that the fourth channel was placed on the extensor carpi ulnaris.

The application Myo EMG Visualizer 2.0, developed by Nicola Corti in 2020 [34], was used to acquire the sEMG signals (Figure 2b). These data were saved in comma-separated file (“.csv”) format, which was sent to a computer via universal serial bus (USB) communication (Figure 2c). The processing steps (machine learning routines) were developed using MATLAB software (Figure 2d).

Data were acquired from 10 healthy volunteers (7 males and 3 females) of 27 ± 2.2 years old. The experimental procedures followed the Ethical Committee for Research involving Humans from Federal University of Technology—Paraná (5,038,447). The 14 gestures in Figure 1 were acquired from the volunteers, following a rhythm of 30 beats per minute guided by a metronome. The volunteers held a neutral position between each gesture (without performing any movement). The acquisition sequence was the same as presented in Figure 1, obtained 30 acquisitions for volunteers. Aiming to avoid muscle fatigue, each acquisition was performed with one minute of pause between gestures. In total, the database was compounded by 420 gestures from each volunteer, equivalent to 4200 gestures.

The segmentation step is illustrated in Figure 3a. An algorithm for automatic segmentation was employed, which was developed in an our previous work, presented in [35]. Its use was to avoid the visual inspection that can be laborious for sEMG processing. It was chosen over other identification methods because it presents results higher than similar algorithms for sEMG signal in pattern recognition processes, such as the double threshold onset method [35]. The used method does not need adjustment of window size after the signal identification, which has an impact in accuracy for sEMG signal classification [36]. Our algorithm can identify the start and the end of an sEMG signal for each gesture, which was suitable for this offline application.

The used segmentation algorithm was divided into three stages: signal pre-processing, threshold delimitation for onset and offset detection, and signal windowing. Some parameters were inserted in the pre-processing stage, as presented in Table 1. Initial parameters were the number of gestures in each acquisition batch, the number of volunteers, the number of acquisitions by volunteers, and the estimated time in each sEMG signal. A parameter named assertiveness rate was applied in this algorithm to evaluate if the identified signals were similar to the identified length of the gesture. The assertiveness rate (ART) was obtained based on a variation coefficient [37], as in Equation (Equation 1):(1)$$ART=100\frac{1-s}{x},$$
where *s* is the standard deviation and *x* is the mean of signal distribution; it was multiplied by 100, an equivalent to a variation coefficient.

The last parameter in Table 1 is the limit for threshold value that is expressed by squared acquisition units (aq·u^2^) which were extracted from the smoothed signal after applying a root mean square (RMS) operator.

Thus, fewer noise channels were identified to be employed in the signal identification process. A histogram for each channel was extracted, and the four channels with the smallest standard deviation on the histogram were selected to smooth sEMG signals and delimit the onsets and offsets of the signal. The standard deviation demonstrates the dispersion on a dataset. Accordingly, the higher the standard deviation, the higher the muscular activity of the signal when compared with the rest moments presented on the acquisition. After selecting the four most relevant channels, their values were summed and smoothed by an RMS operator with a moving average window of 500 ms.

The second algorithm stage was the test of thresholds, progressing from 0 to the average signal value, which were incremented in step 1 to identify the 14 gestures with their respective onsets and offsets. If the algorithm identified fewer than 14 gestures, it increased the threshold and tested again. If it identified more than 14 gestures, the integral value of these gestures was calculated, and it selected the gestures with higher values because they likely indicated more muscular activity and probability of being one of the desired gestures. After finding the 14 gestures, the obtained signals were evaluated according to the ART. Finally, the third stage was the signal windowing, performed for all volunteers and acquisitions. Data were normalized on a range of −1 and 1 before feature extraction.

In total, 19 features were extracted from sEMG signals: AR, CC, DASDV, HIST, IAV, LD, LS, MAV, MFL, MNP, MSR, RMS, SampEn, SSC, TTP, VAR, WAMP, WL, and ZC. This process is exemplified in Figure 3b. Frequency domain features (TTP and MNP) were extracted after the signal passed through the Fast Fourier Transform algorithm. Some features required parameters as follows:AR: 4th-, 6th-, 9th-, and 15th-order coefficients;CC: 4th- and 9th-order coefficients;HIST: Nine bins;LS: Second moment;SampEn: Dimension = 2 and *R* = 0.2.

Furthermore, the features were grouped into 13 sets, as listed in Table 2. These groups were formed based on previous and related works for gesture recognition using sEMG signals.

Continuing Figure 3c, six classifiers were employed: Linear Discriminant Analysis (LDA), Quadratic Discriminant Analysis (QDA), k-Nearest Neighbour (KNN), Random Forest (RF), Support Vector Machine (SVM), and Multilayer Perceptron (MLP) neural network [45,46,47,48,49]. MATLAB was used to implement these classifiers, and the LibSVM library, which applies a one-vs.-one method, was applied for SVM [50]. Cross-validation was employed to parameterize these algorithms, which were defined as follows:KNN: Euclidean distance and 1 nearest neighbour;RF: Thirty trees;SVM: Radial basis function, *C* = 10, Gaussian size = 0.1, one-versus-one method;MLP: Learning rate = 0.0025, 1 hidden layer with 30 neurons, hyperbolic tangent function on hidden layer and logistic function on output layer, training by backpropagation method, stop criterion based on number of epochs, and precision of mean squared error (10^−7^).

The k-fold cross-validation was employed in pattern recognition, with 10 folds applied in this work. As demonstrated in Figure 3c, the classifiers and the feature sets were tested, aiming to find the best combination between them. In this test, all the database was utilized.

Furthermore, three additional analyses were employed, as shown in Figure 3d. The first was the use of ensembles to improve the accuracy of classification. This was performed by analyzing the three classifiers with the highest hit rate in each fold by manual selection. The majority vote removed the final decision for each class. The voting system of the ensemble was developed by taking into account the following criterion: if two or three classifiers resulted in recognition of the same gesture, this gesture was defined as correct; if each classifier provided a different gesture as a response, the right gesture was chosen following the classifier with the highest hit rate. Aiming to reduce the computational cost, the ensemble was implemented with the feature set with the best result on classification (Figure 3c).

The second analysis (Figure 3d) was performed to verify the impact of the pattern recognition process on the gestures. The last test in Figure 3d was the evaluation of the volunteers’ impact on the database. Two approaches were analyzed: the first verifying the results provided from the ten volunteers’ data, and the second for the volunteers individually.

Statistical analyses were performed after each test, aiming to verify the contribution for each step. A Friedman test was applied to the distributions obtained by the classifiers in each test. As the Friedman test is a non-parametric test, it was used to verify the null hypothesis (i.e., there is a difference in the classification techniques in the different scenarios) with an confidence interval of 5% (*p*-value < 0.05). For some instances, a Tukey post hoc multiple test was performed to verify similar distributions among the groups.

## 4. Results

Before presenting the results of classification, Figure 4 shows the shape of the acquired signals for the 14 SIS gestures during one recording. It is possible to note that some channels have a high definition in separating signal and noise, such as channels 1, 7, and 8, while others present noises mixed with the signal, such as channels 4 and 5. This behavior happens due to the placement of the electrodes in the equidistant shape in armband format, where some acquisition channels are more required during the gesture performance. However, it is visible that the sEMG signals present different patterns for each gesture class.

Using the automatic segmentation algorithm previously mentioned, the onsets and offsets were identified to allow feature extraction. Figure 5 exhibits one example from a record, in which the dashed red line is the threshold for signal identification, the green and orange lines represent the identified limits of the gestures (onsets and offsets, respectively), and the resultant is the smoothed signal by the moving average window by RMS operator. The windows show different values due to the instants of onset and offset found for the algorithm. This is not a problem for this application because the processing is suitable for offline applications, in which our objective is to verify if there is the possibility of a pattern recognition process for SIS gestures. After segmentation, the feature extraction step was employed, considering the features presented in Table 2.

### 4.1. Combination of Feature Sets and Classifiers

As the first experiment in classification was the influence of feature sets with pattern recognition methods, Figure 6 presents the accuracy considering the samples from all databases, in which (a) presents the performance for each classifier considering the chosen feature sets and (b) shows the influence of each set as a function of the algorithms. Observe that RF, MLP, QDA, and KNN classifiers presented similar accuracy.

At the top in Figure 6a, the results of a Tukey post hoc test from the Friedman test are presented for the six classifiers, in which *a* is related to RF, *b* for SVM, *c* for MLP, *d* for LDA, *e* for QDA, and *f* for KNN. A comparison was performed to test if the groups had similar distributions (i.e., *p* > 0.05), shown in the figure by the letters over each column. The LDA classifier was the only algorithm that did not present a similar distribution to the others. RF presented similar distributions to QDA and KNN, while SVM and MLP also showed similar accuracies.

Figure 6b illustrates the impact of feature sets in classifications, which demonstrated that each feature set presented a different accuracy for the classifiers. G1, G3, G5, G6, G8, G9, G11, and G13 are more suitable for SIS gesture recognition than the others. Extracting the averages for all classifiers, Figure 6b summarizes the accuracies for all feature sets and detaches that G13 presents the best precision. It is important to note that G2, G4, G7, and G10 have the same distribution. Even though these combinations were performed, the highest accuracy among the 14 gestures was 0.76 for the SVM and G13 combination, so it was the feature set chosen for the subsequent analysis. Thus, the three best classifiers for this feature set were used for ensemble analysis: SVM, KNN, and RF.

### 4.2. Ensemble and Gesture Analysis

Figure 7 shows the accuracies considering the best feature set (G13) for all classifiers. The entire database (including the 10 volunteers) was used to classify the SIS gestures. With these data, the ensemble classifiers were constructed using the aforementioned logic, with Ens_m being the ensemble with manual selection and Ens_a with automatic selection. The three best classifiers in the G13 feature set (SVM, KNN, and RF) were used for this analysis. On the other hand, for the automatic ensemble, the results of all classifiers were evaluated. In this step, the maximum accuracies reached close to 80% in terms of hit rate.

When analyzing Figure 6a, it is possible to verify that there is no difference (statistical or numerical) in the performance of the evaluated methods. Additionally, the normalized execution time for the slowest case was analyzed, presenting, in ascending order: Ens_a = 1 (the slowest); MLP = 0.91; Ens_m = 0.09; RF = 0.08; SVM = 0.06; KNN = 0.02; QDA = 0.02; and LDA = 0.006 (the fastest).

One can note, in Figure 7, that some distributions are similar, such as SVM, RF, KNN, and the two ensemble methods. Aiming to verify the similarity of these distributions, the statistical Friedman test was performed, which demonstrated a difference between the classes (with *p* < 0.05). The differences between the distributions were visualized by the Tukey post hoc test. Similar distributions are indicated with the letters at the top of the boxplot graph in Figure 7 (i.e., where the *p*-value was higher than 0.05). Comparing the ensembles, RF, SVM, and KNN, they demonstrated similar distribution, with the two ensemble methods being statistically similar. Concerning accuracies, both ensembles presented median hit rates of 76%, similar to the SVM classifier.

Confusion matrices were built with the results of SVM and ensembles, as seen in Figure 8. Furthermore, Table 3 lists the accuracies for the best and worst gestures (in numeric format) for these three techniques. Some gestures were correctly classified, such as Medical Thread on a Spool, Medical Thread Loose, Hemostatic Forceps, Scissors, and Compress, which achieved 100% hit rates for ensembles. However, gestures such as Plier Backaus, Farbeuf Retractor, Needle Holder, Allis Clamp, Anatomical Tweezers, and Rat’s Tooth Forceps presented the worst hit rates. Regarding these last three gestures, one can note that most misclassified samples happened among these three classes. In Table 3, it can be seen that these gestures reached about 50% of the hit rate. Analyzing Table 3, the ensemble behavior plays a role in intensifying the results. If the gesture is more easily classified, the ensemble aids in the decision-making, as with the best-recognized gestures. On the other hand, the other classifier results negatively influence the results, presenting a slight drop in the accuracy, as occurs for the gestures Needle Holder, Anatomical Tweezers, and Rat’s Tooth Forceps.

### 4.3. Volunteers’ Analysis

The last analysis was the subject evaluation, using each volunteer’s data to build one classifier. In this step, the data were used for individual training and test stages. Figure 9a presents the distribution for the used classifiers (RF, SVM, MLP, LDA, QDA, KNN, Ens_m, and Ens_a, also identified by the letters a to h, respectively, which are used to indicate, at the top of the graph, the similarity to each classifier). They showed a different behavior than the previous analysis. It is notable that the accuracies increased between 10 and 20% compared with all training and test databases. Even though SVM and KNN showed similar distributions (as can be seen at the top of Figure 9a), the MLP presented the highest median of accuracy, followed by the individual classification methods. Verifying the ensemble methods, automatic search presented a distribution similar to the MLP and reached an accuracy median of 0.88. The differences among the volunteers are presented in Figure 9b for the best individual classifier (MLP) and ensemble method (automatic search), detailing the results for each individual. The classification responds differently for each person, with volunteers 4, 8, and 10 having values higher than 0.9, whilst 2 and 7 could not reach 0.85 for all SIS gestures.

The increasing use of individual data is better detailed in the confusion matrix presented in Figure 10, with the best and worst recognized gestures listed in Table 4. The same behavior was observed in Figure 8 with increased accuracy in cases where gestures were misclassified, such as gestures 11, 12, and 13. This turned out to be a difference of 20 to 30 percentage points. The most significant drop in the success rates occurred with gesture 1 (Compress), which had already presented 100% accuracy in the tests with the whole database for the Ens_a. However, the other gestures increased their accuracy, indicating a gain for the classification process.

## 5. Discussion

After acquiring the SIS gesture data using a wearable device (Myo) and creating the database, the pre-processing steps were analyzed, starting with segmentation using an automatic algorithm. This was previously tested in [35] for a prosthetic task, was found to be suitable in this investigation. Additionally, it could be adapted for online applications, in which the offset time could be suppressed, and overlap segmentation could be applied to improve results [36,51].

Regarding the distributions in Figure 6a, one can note that the similarity observed for RF, QDA, and KNN is reflected in the feature sets because, in some groups (such as G1, G3, G5, G6, and G9), classifiers had hit rates above 50%. The predilection of some classifiers for feature sets is a behavior that occurs through the handcraft feature extraction process. This effect is visualized by the SVM classifier, which, although it presented the lowest hit rates, came out with equal and even better performance than the other classifiers for groups such as G11 and G13. Classifiers having different accuracies for feature sets is a pattern found in similar sEMG works [52,53,54] and is also visualized in the SIS recognition application.

The classifiers also presented different behavior among them. Classifiers such as LDA have a fast and straightforward training process; however, they require normal distributions in features to provide high accuracies [55]. QDA has a more complex process than LDA, allowing quadratic representation of the data, which could be seen in KNN’s presented accuracies (Figure 6a), with values higher than LDA. KNN, KNN, and MLP showed similar results, but their application has some differences. KNN needs to calculate the distance between each pair of samples, which could be costly with a dataset with several examples and input features; RF did not present data interpretability and could reach an overfitted solution [56,57]. MLP also could reach overfitting, but it can be applied in complex non-linear tasks such as an sEMG classification application. Lastly, SVM depends on the initial kernel, and computational cost depends on the number of input features, which could turn into a complex application [56,57]. However, as can be seen in Figure 6, the SVM classifier reached accuracies above 60% for G11 and G13 feature sets, which demonstrates that it is dependent on features and the selected kernel (such as the radial basis function, used in this work).

The observed feature sets G11 and G13 have few features and presented the best performance for devices, with a 200 samples/second sampling rate provided for the device (Myo armband) [25,41,58]. In addition, groups with complex features that demand a very high output number, such as AR6 and HIST, could be more effective in this type of application. For example, AR6 returns 6 values in its output, while HIST has 9 values multiplied for the channels, reaching 48 and 72 for the 8 armband channels, respectively.

Concerning the insertion of ensemble methods for all databases, they showed similar values to the separate classifiers. As the distributions showed similar values in Figure 7, their use is not justified for offline applications. Moreover, compared with the confusion matrices in Figure 8 and Table 3, these methods intensify the decision-making of the classifiers with high accuracy, decreasing the possibility of false positives. Considering all the databases, no differences (statistical or numeric) were observed in performance, but were observed in execution time, in which the ensemble with manual search provided an execution time up to 11 times smaller than the ensemble with automatic search.

The use of one database, compounded by *N* subjects, for testing and training is not the best strategy for SIS gestures classification. As seen in Section 4.3, using the data from each volunteer to train the pattern recognition model was more efficient to classify these patterns. This is more relevant to real applications. A physician or user of this technology can train the device to acquire and process these signals and patterns. Several reasons support this premise. First is the sEMG signals, which, like any biopotential signal, present a noisy and stochastic nature. Second, although the positioning of the device (and, consequently, the electrodes) is the same for each volunteer, differences in electrode placement could affect the acquired signals. Third, it was noted that this use helped the pattern recognition process, especially for the gestures with higher confusion, exemplified by Figure 8 and Figure 10. Thus, this suggests that, in practical use, the physician or user should submit their data for training before the test. This also suggests that a generalist approach, in which the user would use the equipment without prior training, can be investigated in the future.

Related to the chosen gestures, in all stages of processing, Compress (Figure 1-1), Medical Thread on a Spool (Figure 1-2), Medical Thread Loose (Figure 1-3), Hemostatic Forceps (Figure 1-5), and Kelly Hemostatic Forceps (Figure 1-6) presented the best hit rates. On the opposite side, the gestures that were frequently misclassified were Bistouri (Figure 1-8), Needle Holder (Figure 1-9), Allis Clamp (Figure 1-11), Anatomic Tweezers (Figure 1-12), and Rat’s Tooth Forceps (Figure 1-13). One reason for some different performances is related to the nature of the gestures: some are more different due to hand configuration and movement. For example, Bistouri and Needle Holder are similar, explaining their confusion, as are Allis Clamp, Anatomic Tweezers, and Rat’s Tooth Forceps. Decreasing the number of gestures could help the classifiers to provide better performance. However, this could mean a less generalist version of a system for SIS gesture classification.

### Comparison with Related Works

Table 5 shows the related works involving the identification of surgical instruments and analysis of SIS gestures, as mentioned in Section 1. Vision- and image-based systems are the most developed systems, in which cameras or sensor optics are applied. The most common application in this area is the identification of surgical instruments, in which algorithms based on deep learning are used, presenting limitations due to lighting, material, viewing angle, or the similarity of the instruments [11,12]. Some only treat the execution of a surgical activity (e.g., suture) [13]. Despite the fact that vision systems can classify this type of gesture, other solutions with sensors can be developed, such as gloves and bracelets, based on inertial sensors or muscle activation.

Focusing on works that apply only sEMG signals, refs [16,22] are the only works that employ this kind of signal for SIS identification. Comparing with these two references, our work presents the same acquisition device but more volunteers, gestures, classifiers, and feature selection. In addition, the feasibility of using ensemble methods was verified for this application. One of the results obtained from [16] is that the number of volunteers used in the built model influences the performance of the classifier. The increase in data in the database reduced the precision of algorithms. Moreover, the gestures chosen by [16] did not present a high similarity to gestures chosen in this work, such as Allis Clamp, Anatomic Tweezers, and Rat’s Tooth Forceps, which made the separation difficult for the pattern recognition algorithm. Moreover, the authors performed an attribute selection for the classification, verifying the influence of the channel and the features.

On the other hand, reference [22] used 10 gestures, which were not SIS gestures, and found differences between them. Although they were not SIS gestures, the authors aimed to verify hand pose recognition in robot teleoperation. The authors transformed the SIS gesture recognition problem into an sEMG gesture recognition task. Regarding data processing, the most significant difference with our work is the development of the convolutional neural network (CNN) classifier, which demonstrated a performance of 88.5% accuracy. In this work, we obtained a similar result using the individual database for 14 gestures without a complex classifier algorithm. Moreover, a deep learning approach needs a large amount of data to avoid overfitting, nad has a large computational cost, meaning the training stage needs dedicated processing hardware. Thus, in our application, aiming to embed the processing strategy in a wearable and mobile device, traditional machine learning techniques are adequate; this further owes to the fact that the database used is small, allowing training with this dataset [59,60]. This solution also allows execution in a wearable device that has computational constraints, requiring less computational cost than CNN strategies and a reduced database compared to a deep Learning approach, for which particular hardware devices would be needed.

## 6. Conclusions

In this study, the feasibility of recognition of SIS gestures for telesurgery or robotic surgery using only sEMG was performed. Systems based on sEMG have the advantage of low sensitivity to environmental noise, such as lighting or metallic shine of materials and shadows for systems based on image recognition; or audible alarms from devices, conversations between team members, or other sound sources for systems based on speech recognition.

The system demonstrated that, for offline applications, that is, with the primary objective of studying data processing strategies and pattern classification for SIS, it is possible to reach high hit rates when considering the data for each volunteer. In particular, using a combination of classifiers (ensemble) and a feature set that employs attributes from sEMG such as MSR, LS, WAMP, and MFL, it could reach accuracies close to 90% (on average) for 14 SIS gestures. For some subjects, the accuracies were higher than 90%. The errors in classification were observed among similar gestures, which were improved using the ensemble methods. Comparing with related works, our paper presents several advantages: less sensitivity to environmental noise due to use of sEMG signals, data acquisition being performed for a larger number of volunteers, and signals being acquired for a larger number of gestures. In this work, we obtained similar results using a larger number of gestures (14), without using a complex classifier algorithm, when compared to the few works listed in the state of the art, which used a smaller number of gestures. (The one that used the most had 10, but these were not SIS gestures; the two works with 5 SIS gestures were the next highest.)

As this work is an initial study, the results could have reflected the use of offline processing techniques. In an application in an operating room, priority must be given to using an online device, which is one of the main future works to be developed. Subsequently, increasing the database of acquired gestures by acquiring more samples of more gestures, mainly similar gestures, from more volunteers should be obtained to improve the pattern recognition techniques. Regarding new techniques, as the related works used CNNs and deep learning, inserting them in the processing of these signals should be studied.

Finally, it is also equally important to verify the potential benefits of the insertion of other sensor technologies, such as inertial sensors, flexible sensors, and image-based processing, to help in the identification of similar gestures, where they are differentiated by the position of the hand or by the execution in movement.

## Figures and Tables

**Figure 1 sensors-23-06233-f001:**
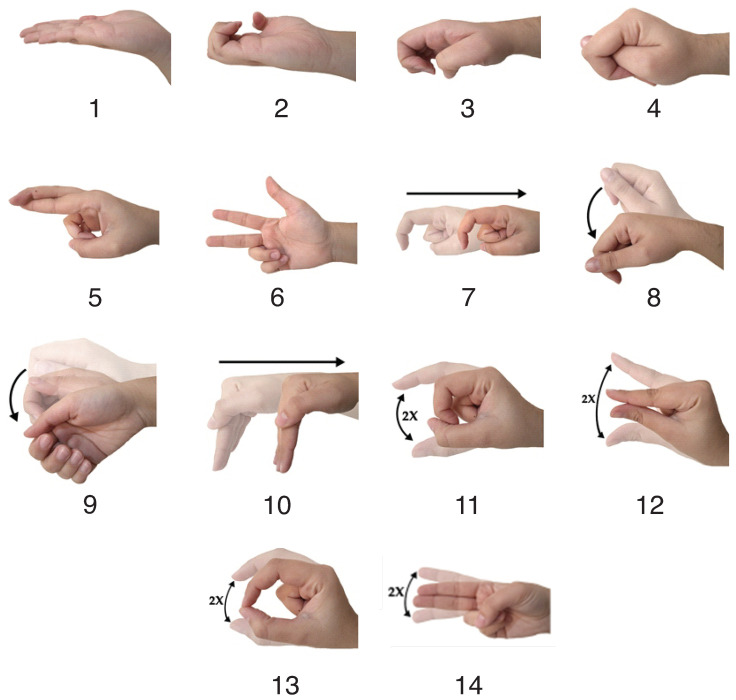
Gestures for SIS used in this work: (**1**) Compress, (**2**) Medical Thread on a Spool, (**3**) Medical Thread Loose, (**4**) Plier Backhaus, (**5**) Hemostatic Forceps, (**6**) Kelly Hemostatic Forceps, (**7**) Farabeuf Retractor, (**8**) Bistouri, (**9**) Needle Holder, (**10**) Valve Doyen, (**11**) Allis Clamp, (**12**) Anatomical Tweezers, (**13**) Rat’s Tooth Forceps, (**14**) Scissors.

**Figure 2 sensors-23-06233-f002:**
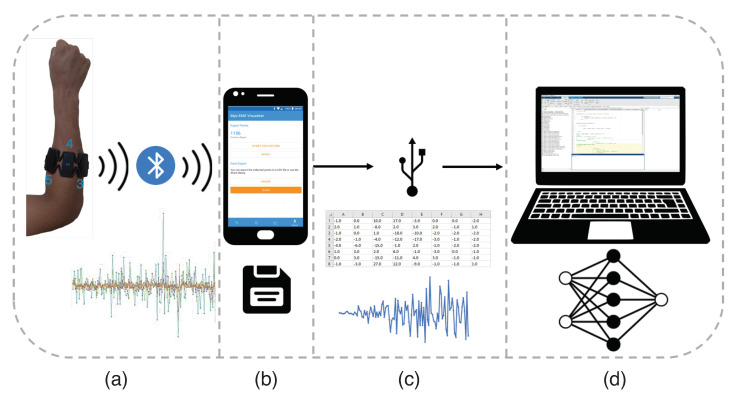
Data acquisition flow. (**a**) sEMG signals are acquired from the Myo armband placed on the right forearm. (**b**) The sEMG signals are sent by Bluetooth from a mobile application. (**c**) The mobile application on the smartphone saves the signal for each acquisition in a “.csv” file, which is sent to a computer via a USB connection. (**d**) On the computer, data are processed using machine learning routines on MATLAB.

**Figure 3 sensors-23-06233-f003:**
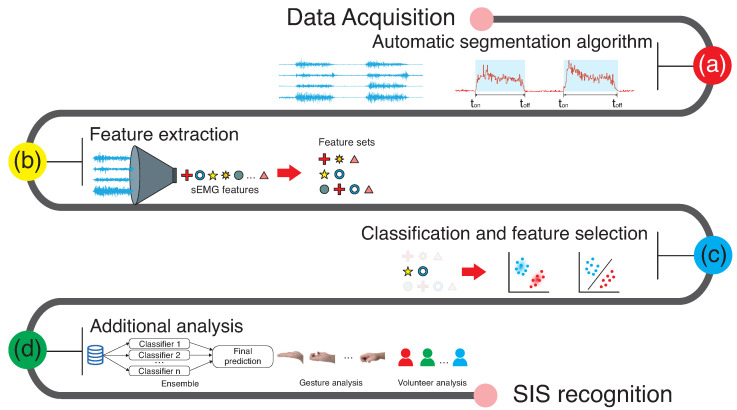
Processing steps followed in this work. (**a**) After data acquisition, sEMG signals were segmented using an automatic routine that identified the start and the end of each sEMG activation. (**b**) Features were extracted from segmented signals, building feature sets based on the literature and previous works. (**c**) Feature sets were tested and selected using the classifiers. (**d**) Some additional analyses were performed, such as combining the classifiers as ensembles, evaluating the gestures, and determining feasibility from the volunteers.

**Figure 4 sensors-23-06233-f004:**
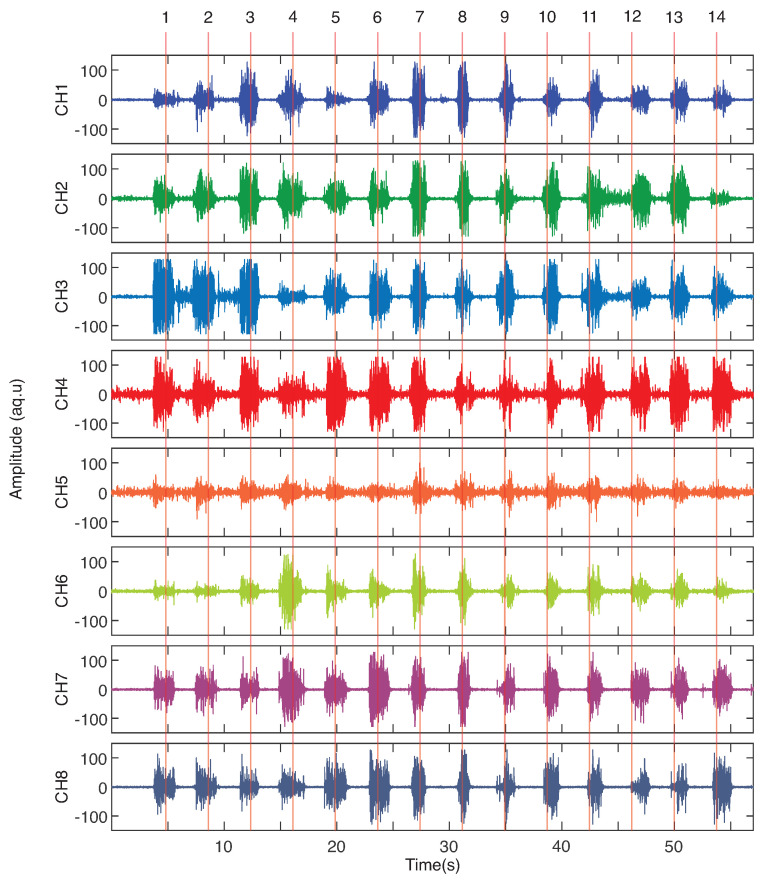
Example of sEMG signals obtained from the chosen 14 SIS. The gestures from (**1**) to (**14**) are the same as presented in Figure 1: (**1**) Compress, (**2**) Medical Thread on a Spool, (**3**) Medical Thread Loose, (**4**) Plier Backhaus, (**5**) Hemostatic Forceps, (**6**) Kelly Hemostatic Forceps, (**7**) Farabeuf Retractor, (**8**) Bistouri, (**9**) Needle Holder, (**10**) Valve Doyen, (**11**) Allis Clamp, (**12**) Anatomical Tweezers, (**13**) Rat’s Tooth Forceps, and (**14**) Scissors.

**Figure 5 sensors-23-06233-f005:**
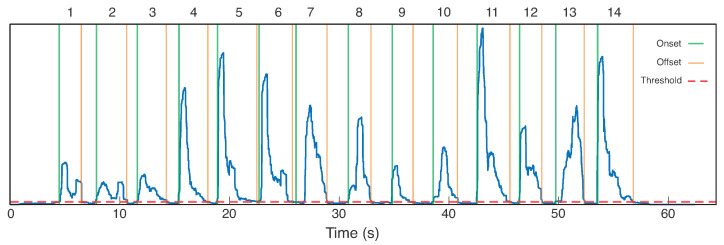
Example of the segmentation applied in this work. The resultant signal is a combination of the four most relevant channels. The threshold signal is used to find the 14 SIS gestures, as demonstrated in Figure 1 and Figure 4 (numbers are highlighted at the top). Moreover, the onset and offsets automatically found by the developed algorithm are presented by the green and orange lines, respectively.

**Figure 6 sensors-23-06233-f006:**
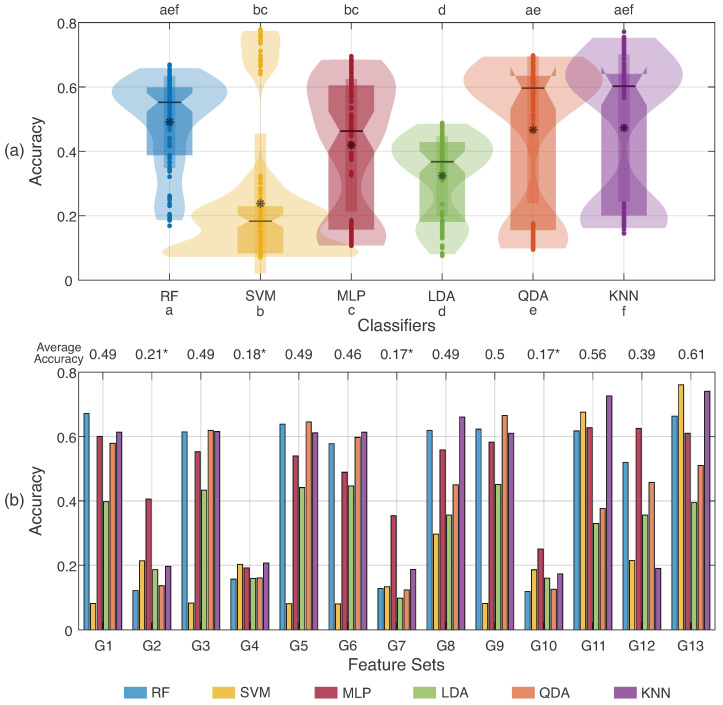
(**a**) Distribution of accuracies for each classifier considering the data from all volunteers. The markers at the top of the graph represent the results for the similar distributions to each classifier using the Tukey post hoc test from Friedman’s test (*p* > 0.05). The letters represent each classifier (a—RF, b—SVM, c—MLP, d—LDA, e—QDA, and f—KNN). (**b**) Hit rates obtained for each feature set from G1 to G13. At the top, the average accuracies for each feature set are presented. The markers (*) indicate feature sets with statistically different distributions, obtained from the Tukey post hoc test from Friedman’s test (*p* > 0.05).

**Figure 7 sensors-23-06233-f007:**
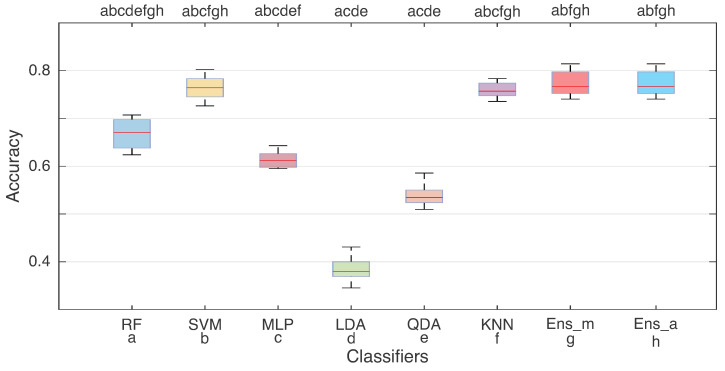
Distributions for the accuracies obtained from the ensemble analysis considering the data from all 10 volunteers from the training and test step. Ens_m is the hit rate from ensemble method for manual search of classifiers, which was the best classifier for the G13 feature set. Ens_a is the results obtained from the ensemble method with automatic search, considering all the classifiers. At the top, the results of the Tukey post hoc test from Friedman’s test are presented (*p* > 0.05). The letters represent each classifier (a—RF, b—SVM, c—MLP, d—LDA, e—QDA, f—KNN, g—Ens_m, and h—Ens_a).

**Figure 8 sensors-23-06233-f008:**
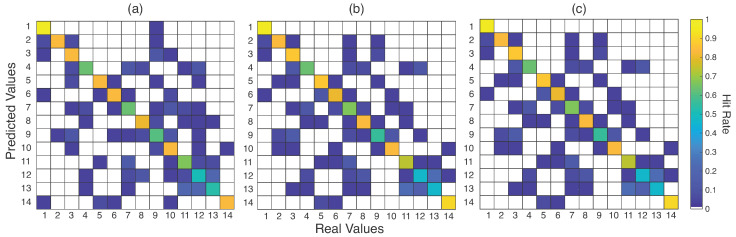
Confusion matrices for the selected 14 SIS gestures (Figure 1) and G13 feature set for SVM classifier (**a**), and for the manual (**b**) and automatic (**c**) ensembles. Null cells represent 0% misclassification. Gestures 1 from 14 refer to the gestures from Figure 1.

**Figure 9 sensors-23-06233-f009:**
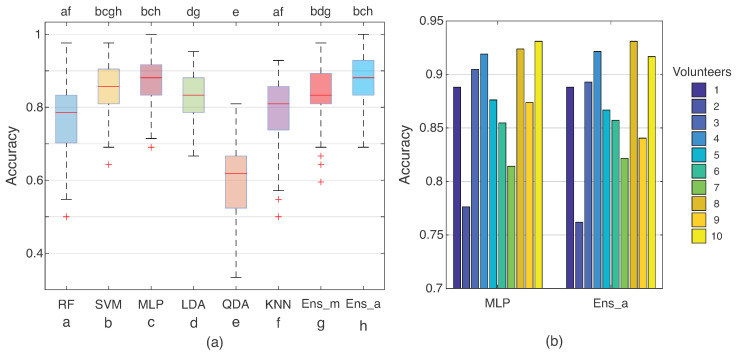
(**a**) Distributions for individual accuracies for the classifiers (using G13 feature set), considering the data from each volunteer for individual training and test steps. At the top, the Tukey post hoc test from Friedman’s test are presented (*p* > 0.05). The letters represent each classifier (a—RF, b—SVM, c—MLP, d—LDA, e—QDA, f—KNN, g—Ens_m, and h—Ens_a). In (**b**), the accuracy obtained from each volunteer for the algorithms with the best accuracies, i.e., MLP and ensemble with automatic decision, are presented.

**Figure 10 sensors-23-06233-f010:**
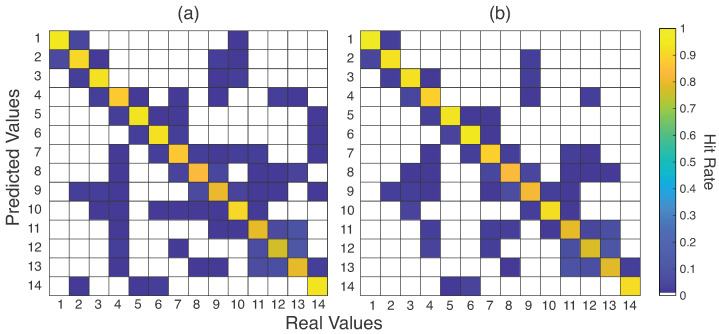
Confusion matrices for (**a**) MPL and (**b**) ensemble with automatic search. Null cells represents less than 1% misclassification. Gestures 1 from 14 refer to the gestures from Figure 1.

**Table 1 sensors-23-06233-t001:** Segmentation parameters.

Parameter	Value
Number of gestures	14
Number of volunteers	10
Acquisitions by volunteer	30
Estimated time for each sEMG activation	2 s
Assertiveness rate	80%
Limit value of amplitude for threshold	500 aq·u^2^

**Table 2 sensors-23-06233-t002:** Feature sets employed in this work.

Feature Set	Features	Reference
G1	MAV, WL, ZC, and SSC	[38]
G2	RMS and AR6	[39]
G3	MAV, WL, ZC, SSC, RMS, and AR6	[39]
G4	AR4 and HIST	[40]
G5	WL, LD, SSC, and AR9	[41,42]
G6	WL, SSC, AR9, and CC9	[41,42]
G7	RMS, VAR, LD, and HIST	[41,42]
G8	WL, RMS, SampEn, and CC4	[43]
G9	AR15, ZC, MAV, RMS, SSC, and WL	[44]
G10	MAV and AR4	[25]
G11	MFL, MSR, WAMP, and LS	[41]
G12	LS, MFL, MSR, WAMP, ZC, RMS, IAV, DASDV, and VAR	[41]
G13	MFL, MNP, TTP, and RMS	[25]

**Table 3 sensors-23-06233-t003:** Hit rates for the best and worst recognized SIS gestures for the confusion matrices in Figure 8, for SVM classifier, manual (**Ens_a**) and automatic (**Ens_m**) ensembles.

Best Recognized Gestures	SVM	Ens_a	Ens_m
1	0.97	1	1
3	0.83	0.87	0.87
5	0.83	0.87	0.87
6	0.83	0.8	0.8
10	0.83	0.83	0.83
**Worst Recognized Gestures**	**SVM**	**Ens_a**	**Ens_m**
4	0.63	0.63	0.63
7	0.63	0.67	0.67
9	0.6	0.57	0.57
12	0.65	0.47	0.47
13	0.53	0.47	0.47

**Table 4 sensors-23-06233-t004:** Hit rates for the best and worst recognized SIS gestures for the confusion matrices in Figure 10, for MPL and ensemble with automatic search.

Best Recognized Gestures	MLP	Ens_a
1	0.94	0.96
2	0.93	0.95
3	0.93	0.93
5	0.92	0.93
6	0.91	0.95
**Worst Recognized Gestures**	**MLP**	**Ens_a**
8	0.87	0.83
9	0.87	0.81
11	0.81	0.79
12	0.8	0.77
13	0.79	0.79

**Table 5 sensors-23-06233-t005:** Comparison with related works.

Reference	Objective	Techniques	Gestures/Instruments	Results
[11]	Recognition of instruments and their placement	- Kinect sensor - Image classification	- 5 gestures - Numbers representing instruments	System is viable for classification
[12]	Recognition of surgical instruments	- Image classification - CNN classifier structured by DT	- 10 instruments - 5 instruments	10 instruments: 70% 5 instruments: 96%
[13]	Recognition of suturing gestures	- Image classification - Deep learning	-	Gesture presence detection: 88% Gesture identification: 87%
[21]	Recognition of instruments placed on holder	- Optical sensor - Image classification - Robotic manipulation	-	95.6% accuracy
[22]	Gesture recognition	- EMG: Myo Armband™ - Several classifiers, such as SVM, KNN, and convolutional neural networks	10 gestures representing numbers from 1 to 10	88.5% for deep CNN
[16]	Recognition of Surgical Instrument Signaling gestures	- EMG: Myo Armband™ - ANN and DT classifiers	5 instruments and gestures	1 volunteer: 95.2% (ANN) and 95.3% (DT) 5 volunteers: 70.1% (ANN) and 71.2% (DT)
This work	Recognition of Surgical Instrument Signaling gestures	- EMG: Myo Armband™ - Several classifiers and ensembles - Feature selection - Comparison with volunteers	14 SIS gestures	10 volunteers: 76% (SVM) Individual classification: 88% (ensemble method)

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
