# Peer review of "Surgical Instrument Signaling Gesture Recognition Using Surface Electromyography Signals"

_sensors, 2023, doi:10.3390/s23136233_

Round 1
Reviewer 1 Report
This paper developed a processing routine for telesurgery or robotic surgery applications. A database of 14 selected SIS gestures was created from 10 volunteers. Segmentation, feature extraction, feature selection, and classification techniques were employed, and multiple parameters were evaluated. The system was tested offline and showed positive results for all databases and individual volunteers. Two ensemble techniques helped distinguish the sEMG signals for the 14 SIS gestures. The Support Vector Machine classifier achieved an accuracy of 76% for all databases and 88% for individual volunteer analysis. However, I have the following comments for the authors to consider.
(1) In the Introduction, the author mentions that the pattern recognition of SIS-based systems is highly dependent on environmental factors. Therefore, these factors should be considered in this paper's data collection and experimental analysis.
(2) In Methodology, the authors used an automatic segmentation algorithm to segment the images. I wonder if the authors have tried other segmentation algorithms and suggest that the authors explain the reason for using this algorithm.
(3) Figure 6(a) presents the performance of each classifier considering the chosen feature sets. I would like to know if different feature sets affect the results and suggest the authors briefly state.
(4) In 5.1, I found that the method proposed in this paper is similar to the existing methods. Therefore, it is recommended to analyze the advantages of the proposed method in this paper.
(5) There is a typo in the fifth paragraph of the Discussion.
Some grammatical and tense errors and sentence structures are very strange, but it does not affect understanding.
Author Response
Dear Reviewer, thank you for your valuable comments. You will find the answers to specific questions in the attached document. Thank you in advance for the opportunity to review and improve the dissemination of our work.

Reviewer 2 Report
The authors used sEMG signals acquired from the Myo armband of 10 volunteers to evaluate performance of different combinations of features and classifiers for classification of 14 SIS gestures. The results were 76% of accuracy for the SVM classifier for all databases and 88% for analyzing the volunteers individually.
The proposed manuscript contains some interesting ideas. However, I have some concerns with this work and how it has been written.
1) Why should the authors used smartphone to save the signal and then sent to a computer? If considering in a surgical procedure, it may not be suitable.
2) The experiments need a more elaborate methodical, mathematical description, a thorough presentation and discussion of the achieved results. For example, how to do the preprocessing? (Since the authors discussed its feasibility in future online applications, they should explain the segmentation steps but not just said it is presented in [32].) How to get one accuracy by one-versus-one method in SVM? How to do the statistical analysis?
3) The authors said” Considering all the databases, it was not seen in these methods, no differences (statistical or numeric) in performance but only in execution time, in which the ensemble with manual search provided an execution time four times smaller than the ensemble with automatic search”. However, there was no results about time information in this manuscript.
4) The author may make a good analysis in terms of data collection and process, but the algorithm-based novelty is not discussed wisely.
5) Please check some sentences, such as “The differences between the distributions were visualized by the Tukey post-hoc test (p>0.05)”, “As seen in Section ??, using the data from each volunteer to build the classification model is more efficient for classifying these patterns”.
Author Response
Dear Reviewer 1, thank you for your valuable comments. You will find the answers to specific questions in the attached document. Thank you in advance for the opportunity to review and improve the dissemination of our work.

Round 2
Reviewer 1 Report
The authors have made detailed revisions to the comments already made. However, I have some new comments for the authors to consider.
(1) In Table 5, the author compared the results of this work with previous work, but I still need to find a significant improvement. Especially compared with the deep learning method, the method in this paper has no advantage.
(2) The paper could benefit from a more thorough discussion of the results. For example, it could discuss why certain classifiers performed better than others, why the accuracy increased when using individual data, and the implications of these results.
Author Response
Dear reviewer We appreciate your comment. For sEMG signals involving SIS gestures, no model with deep learning has yet been applied. As discussed in section 5.1, works that apply deep learning use gestures that will symbolize surgical instruments, but are not (as in the case of the work by Qi et al. 2021) or that use images of instruments for their recognition in image without using sEMG. In this case, we are one of the first works that explore a large number of algorithms and techniques for the use of sEMG in this type of application. For this reason, we insert in future works the comparison with models of convolutional neural networks (deep learning).
In addition, for a deep learning model to be successful and generalizable, it is necessary to have a large database, which is not the case in this work. It is necessary to increase the number of samples to justify the use of deep learning models. Another factor is that these models tend to require a high computational cost, being processed in computers or systems with dedicated processing boards. In our machine learning models, we achieved similar results using deep learning. Comparing our work with the closest one that uses deep learning (Qi et al. 2021), our obtained accuracy was 88% and that of Qi et al. 2021, 88.5%. In addition, our model presented 14 gestures, while the model of these authors, 10 gestures that are not SIS gestures. We make these points discussed here more explicit at the end of section 5.1.

Reviewer 2 Report
Appreciate the efforts in revising the manuscript. However, I still have a few concerns with this work.
1) Please clearly explain the main purpose of comparing different classifiers and the use of Friedman test. Also, is there any meaning in the comparison of feature sets (just means using few features?).
2) It would be better to have more figures for details except of Figure 3.
Author Response
Answer1 : We would like to thank the revisor for the comment. We explained with more details the use of Friedman test in the end of Methodology.
"Statistical analyzes were performed after each test, aiming to verify the contribution for each step. Friedman test was applied for the distributions obtained by the classifiers in each test. As the Friedman is a non-parametric test, it was used to verify if the null hypothesis that there is a difference in the classification techniques in the different scenarios with an interval of confidence of 5% (p-value $<$ 0.05). For some instances, a Tukey post-hoc multiple test was performed to verify similar distributions among the groups."
Regarding the feature set, this is because each application involving sEMG signal processing has a more appropriate feature set. The comparison of several feature sets for sEMG problems is a necessary action, and it helps in choosing a set that has high performance and may have reduced computational effort for applications involving online processing. In this work, we used several sets of features because, as the application involving SIS gestures is still new, there is no more appropriate group of features to solve this type of problem.
includes the images and the steps that allow reproduction. In addition, we
have inserted the availability of the codes and database at the end of the
article, which can be done by request to the corresponding author. Regarding
the images, we believe that we covered the entire scope of the work, namely:
Figure 1 - The gestures to be classified;
Figure 2 - The data acquisition method;
Figure 3 - The proposed analysis methodology;
Figure 4 - Example of acquired signals;
Figure 5 - Example of how the segmentation process works;
Figure 6 - Results for all classifiers and feature groups considering the
entire database;
Figure 7 - Results with ensemble methods;
Figure 8 - Confusion matrices for the entire database;
Figure 9 - Accuracy with separate training and testing for each individual;
Figure 10 - Confusion matrices for analyzing separate training and testing
for each individual.
